# History, Rats, Fleas, and Opossums: The Ascendency of Flea-Borne Typhus in the United States, 1910–1944

**DOI:** 10.3390/tropicalmed5010037

**Published:** 2020-03-01

**Authors:** Gregory M. Anstead

**Affiliations:** 1Medical Service, South Texas Veterans Health Care System, San Antonio, TX 78229, USA; anstead@uthscsa.edu; Tel.: +210-567-4666; Fax: +210-567-4670; 2Department of Medicine, University of Texas Health Science Center at San Antonio, San Antonio, TX 78229, USA

**Keywords:** rats, fleas, *Rickettsa typhi*, *Rickettsia felis*, murine typhus, insecticide, rodenticide

## Abstract

Flea-borne typhus, due to *Rickettsia typhi* and *Rickettsia felis*, is an infection causing fever, headache, rash, hepatitis, thrombocytopenia, and diverse organ manifestations. Although most cases are self-limited, 26%–28% have complications and up to one-third require intensive care. Flea-borne typhus was recognized as an illness similar to epidemic typhus, but having a milder course, in the Southeastern United States and TX from 1913 into the 1920s. Kenneth Maxcy of the US Public Health Service (USPHS) first described the illness in detail and proposed a rodent reservoir and an arthropod vector. Other investigators of the USPHS (Eugene Dyer, Adolph Rumreich, Lucius Badger, Elmer Ceder, William Workman, and George Brigham) determined that the brown and black rats were reservoirs and various species of fleas, especially the Oriental rat flea, were the vectors. The disease was recognized as a health concern in the Southern United States in the 1920s and an increasing number of cases were observed in the 1930s and 1940s, with about 42,000 cases reported between 1931–1946. Attempts to control the disease in the 1930s by fumigation and rat proofing and extermination were unsuccessful. The dramatic increase in the number of cases from 1930 through 1944 was due to: the diversification of Southern agriculture away from cotton; the displacement of the smaller black rat by the larger brown rat in many areas; poor housing conditions during the Great Depression and World War II; and shortages of effective rodenticides and insecticides during World War II.

## 1. Introduction

Flea-borne typhus (FBT), also known as murine typhus, is an infection caused by *Rickettsia typhi* and *R. felis*. The infection is transmitted to humans by the inoculation of a bite site, a skin abrasion, or mucous membranes with feces from fleas infected with these rickettsiae or by a flea bite [1,2]. Flea-borne typhus is the most prevalent and widely distributed rickettsial infection [1,3]. In the United States, FBT is now relatively uncommon, but foci still exist in TX, HI, and CA [4]. However, 80 years ago, the disease was common throughout the Southeastern United States, with 5401 cases reported in 1944 [5]. Historically, over 90% of the FBT cases in the United States were reported from eight Southern states [1]. In the new millennium, as we reflect upon the achievements in public health in the United States in the last century [6,7], the control of FBT is one of those successes. To quote Dumler and coworkers: “Few zoonotic diseases have been curtailed through control of vector and vector-host populations as well as murine typhus.” [8]. Nevertheless, the reported incidence of FBT in TX and CA has more than doubled over the last decade [9,10]. It is instructive to review the history of the ascendency, decline, and resurgence of FBT in the United States because it exemplifies the influence of social conditions, public policy, and technology on the epidemiology of a vector-borne disease. The title of this paper pays homage to the classic book on epidemic typhus “Rats, Lice, and History,” by microbiologist Hans Zinsser [11]. The thesis of Zinsser’s book is that epidemic typhus, due to louse-borne *R. prowazekii*, shaped European and Latin American history. The opposite argument will be made for FBT in the United States; historical forces led to the social and economic conditions that promoted an epidemic of FBT in the Southern United States from 1930 to 1945, but also fostered the technologies and social change that facilitated its control. This public health epic is divided into three parts. Part I will include: a clinical overview of FBT; an account of the discovery of the flea vectors and the rat reservoirs; and an examination of the historical and social factors that led to a progressive increase in the number of FBT cases in the United States from 1920 to 1944. Opossums are mentioned in the title, but they do not play a significant role in this initial time period; their importance will be revealed in the subsequent parts. Part II (the decline) will describe: the innovations in insecticide and rodenticide technology occurring during and after World War II; the public health programs instituted for typhus and rodent control in the post-war period; and the improvement in social conditions in the endemic areas after World War II. Part III (the resurgence) will address the change in the epidemiology of FBT from a rodent flea-borne disease to a cat flea-borne disease, with opossums as a major reservoir; the discovery of a second causative bacterium (*R. felis*); and epidemiologic trends in the persistent FBT endemic areas of the United States (TX, CA, and HI).

## 2. Clinical Presentation of Flea-Borne Typhus

The incubation period of FBT ranges from 8–16 days, with a mean of 11 days. Prodromal symptoms start 4–6 days after inoculation, which include headache, backache, malaise, nausea, and arthralgia. In 60% of cases, a shaking chill heralds the onset of the active phase of infection. This is accompanied by a severe frontal headache, which lessens in intensity as the illness progresses [12] but does not resolve until the fever remits [13]. Typically, the chill, headache, fatigue, body aches, and fever all transpire within hours, often with nausea and vomiting. In the first 3 to 4 days of the illness the fever rises progressively, reaching a maximum of 39.4–40.0 °C in adults and 41.1 °C in children. The fever typically lasts for 9–14 days [14,15,16]. The infection tends to run a milder course in children [17]. The rash usually starts on day five after fever onset but may be missed initially because it may be high in axilla or the inner surface of the arms. However, the onset of the rash may be coincident with the fever or appear as late as 8 days after fever onset. The rash spares the face, the palms, and the soles. In one series of 80 patients, a rash occurred in 54%; the erythematous rash was macular in 49%, maculopapular in 29%, papular in 14%, petechial in 6%, and morbilliform in 3% [8].

Tsioutis and colleagues provided an analysis of the presentation of 2074 adult and pediatric patients with FBT [18]. In addition to fever, common symptoms of FBT include: headache (81%); fever and headache (73%); fever and rash (53%); malaise (67%); chills (63%); myalgias (52%); anorexia (48%); back pain (35%); a triad of fever, headache, and rash (35%); cough (27%); arthralgia (28%); nausea and/or vomiting (27%); diarrhea (19%); abdominal pain (18%); sore throat (14%); photophobia (10%) [18]; and confusion (2%–14%) [8,19,20]. Common signs of infection are: rash (48%), hepatomegaly (22%), conjunctivitis (18%), splenomegaly (17%), and lymphadenopathy (13%). Due to the late appearance of the rash or absence of the rash, the clinical triad of fever, headache, and rash is not reliable for the diagnosis of FBT [8,21]. The most common laboratory abnormalities include: elevation of the transaminases (79%), lactate dehydrogenase (73%), alkaline phosphatase (41%), and creatinine kinase (62%); hypoalbuminemia (60%); and hyponatremia (34%). Hematologic findings are thrombocytopenia (42%), anemia (38%), leukopenia (24%), and leukocytosis (18%) [18].

Although FBT is typically an acute self-limited illness, complications occur in about 26%–28% of cases [14,18,20], which can be systemic, neurological, cardiac, pulmonary, gastrointestinal, renal, hematologic, or involve other systems (Table 1). Due to this myriad of presentations, FBT is often initially misdiagnosed, even in endemic areas [8]. In a 2008 series of 33 patients with FBT from Austin, TX, 70% of patients were hospitalized, and even though there were no fatalities, one-third required intensive care [19]. In a series of 80 patients from TX (1980–1987), there were three deaths (3.8%), two from shock and multi-organ failure [8]. Eleven fatal cases of FBT were reported in TX from 1985–2015, out of 3048 cases (0.4% mortality rate) [22]. The low mortality rates reported in these series belie that FBT can nevertheless be a very serious illness, necessitating intensive care unit admission [23] and prolonged convalescence and causing permanent sequelae. Deaths in autopsy cases were ascribed to myocarditis, septic shock, and multi-organ failure (encephalitis, pneumonia, and renal failure) [24,25,26].

## 3. The Vectors of Murine Typhus

More than eleven species of fleas in eight different genera are implicated as vectors of FBT [1,27,28]. Most species of fleas are intermittent feeders that travel from host to host and consume multiple blood meals per generation, which facilitates the dissemination of rickettsiae to multiple hosts [29]. Five flea species are, or have been, significant to the prevalence of FBT in the United States: Oriental rat flea (*Xenopsylla cheopis*), European mouse flea (*Leptopsylla segnis)*, chicken flea, (*Echidnophaga gallinacea)*, Northern rat flea (*Nosopsyllus fasciatus*), and cat flea (*Ctenocephalides felis*). During the period in the United States when rats were the dominant reservoir, the Oriental rat flea was the most abundant vector of FBT [28,30,31]. In a 1927 survey of 5861 fleas collected from 887 brown rats in Savannah, GA, *X. cheopis* comprised 68% of the fleas; *L. segnis*, 14%; *N. fasciatus*, 7%; and *E. gallinacea*, 3% [32]. Of 4898 fleas taken from 1561 rats in Norfolk, VA in 1929, 82% were *X. cheopis* and 17% were *N. fasciatus* [33]. Based on studies conducted in AL, FL, and GA from 1932–1934, 63%–85% of the rats were infested with *X. cheopis* [34]. The importance of the Oriental rat flea as an intermurid vector of FBT was evident in a study from AL in 1943. *Xenopsylla cheopis*, *L. segnis*, and *E. gallinacea* were all recovered from trapped rats, but infestation with *X. cheopis* was the most strongly associated with FBT seropositivity [31]. The Oriental rat flea prefers warm, humid climates (temperatures of 18.3 to 26.7 °C), with humidity levels of ≥ 70% that are favorable for the hatching of its eggs [35]; such conditions prevail in the American Southeast [34].

The European mouse flea is also naturally infected with *R. typhi* in the United States [28]. Despite its moniker, this insect prefers to feed on rats [36]. In the laboratory, *L. segnis* is more effective than *X. cheopis* in transmitting the infection to rats; because rickettsiae are present in the foregut and proventriculus, organisms may be transmitted by its bite. As a semi-sessile flea, *L. segnis* remains attached to a rodent for days while feeding and thereby may acquire a higher inoculum of rickettsiae than rapidly feeding flea species, such as *C. felis* or *X. cheopis* [37,38]. Although *L. segnis* does not commonly bite humans, it is important in the epidemiology of FBT because it spreads rickettsiae from rodent to rodent and it may transmit FBT to humans via its aerosolized feces, which are infective for years [39,40]. The chicken flea feeds on fowl, rodents, dogs, and cats and is more abundant in rural areas [35,41,42]. The female chicken flea stays attached to a single host for prolonged periods, lessening its vectorial significance [35]. *Nosopsyllus fasciatus* does not readily attack humans [36] but may serve as an intermurid vector. The fecally-contaminated pelage of rats serves to transmit the infection to other rats via inhalation or conjunctival inoculation by close contact in a burrow or during mutual grooming [1]. However, cannibalism and fighting are not means of intermurid FBT transmission [43]. The cat flea is currently abundant in areas of TX and CA with FBT and has a propensity to feed not only on humans, but also on dogs, cats, and urban wildlife, such as opossums, skunks, and raccoons [44]. Rats may be parasitized by the cat flea, but are not the preferred hosts [45,46].

Once a flea imbibes a blood meal infected with *R. typhi* or *R. felis*, within hours the rickettsiae enter the midgut epithelial cells, and replication occurs within the cytoplasm. The rickettsiae are released into the gut lumen and excreted in the feces 3–4 days after initial infection, but do not reach sufficient levels in the feces to cause infection of another rat for 6–10 days [1,47]. Some rickettsiae escape the midgut and enter the hemocoel of the infected flea [1] and thus fleas are able to pass both rickettsiae to their progeny by transovarial transmission [1,27,48,49]. Transmission of rickettsiae from flea to flea does not occur sexually or by contact, nor are the larvae infected by feeding on infected eggs or dejecta [27,48]. Fleas infected with *R. typhi* retain the infection for at least 52 days; thus, they serve as both vectors and reservoirs [50]. Unlike the situation with *Yersinia pestis* (the plague bacterium), which is lethal to rats and fleas alike, neither are adversely affected by *R. typhi* or *R. felis* [1,3,29]. The ability of infected fleas to withstand persistent rickettsial infection without harm facilitates pathogen transmission. The maintenance of *R. felis* within infected populations of *C. felis* has been documented for 12 generations without affecting the vector’s fitness. This multi-generational maintenance of rickettsiae in the cat flea population without the need for a reservoir host indicates the importance of transovarial transmission in perpetuating the infection [48].

When there is a high percentage of rats in a population infested with *X. cheopis*, the flea burden per rat increases. If 31%–35% of rats in a population are flea-infested, the average number of fleas/rat is one. At 66%–70% flea infestation, the average flea burden per rat is four. At 76%–80% flea infestation, the average flea burden jumps to eight [34].

## 4. The Reservoirs

Historically, two species of rats served as the main reservoirs of FBT in the United States: the black or roof rat (*Rattus rattus*) and the brown or Norway rat (*Ra. norvegicus*). The black rat is distinguished by its smaller size (maximum size 200 g vs. 400–500 g for the brown rat), prominent eyes and ears, and longer tail. Rats are well known for their prodigious reproductive capacity. They mature quickly (three months), have short gestation periods (three weeks), and have 3–5 litters per year, with 6–10 pups/litter [51]. In the first two decades of the 20th century, the black rat was the most common rat species in the Southern United States [52,53,54], but in the 1930s and 1940s, the brown rat displaced the black rat in many locales [55,56]. By the early 1930s, the brown rat comprised 94% and 99.8% of the rats in Savannah and Atlanta, GA, respectively [57]. Reece reported the brown rat as the predominant rat species in TX in the early 1930s [13]. However, the predominant species in a particular area depends on local conditions and control measures [58]. The specific rat species is important to the epidemiology of FBT because the larger brown rat can harbor about four-times as many fleas as the black rat [59]. Rats serve as amplifying hosts for the rickettsiae; the period of bacteremia in the rat is brief (one to two weeks), but the organisms persist in the brain for months after infection [38].

Both species of rats are omnivorous and fierce competitors for the detritus of humanity. In general, the black rat prefers grocery and poultry stores, barns, corncribs, and grain storage facilities to human dwellings and often inhabits the attics, rafters, and crossbeams of buildings. By contrast, the brown rat tends to occupy basements and ground floors, and burrows under sidewalks and outbuildings [52,60]. Thus, the brown rat is usually easier to destroy [61]. The brown rat, larger and more aggressive, often outcompetes the black rat, but the two species may co-exist in multi-story buildings in which black rats occupy the upper stories and brown rats inhabit the first floor and basement [55]. The brown rat is also more likely to live in a feral state [52]. Flea-borne typhus has been called a “man-made disease” because humans transported rats and their ectoparasites around the world and created ideal environmental conditions for rodents to flourish [28]. However, starting in the 1960s, studies in TX and the Los Angeles, CA suburbs indicated that the classic rodent–flea cycle involving *R. typhi* and *R. felis* was supplanted by a peri-domestic cycle involving opossums, cats, dogs, and the cat flea [62,63,64].

Although the house mouse *Mus musculus* may also harbor *R. typhi*, this rodent is not important in the transmission to humans in the continental United States [29,65]. The number of *X. cheopis* is low on mice compared to rats because the blood of mice is deficient in nutrients required for high fecundity in *X. cheopis* [66]. Due to the persistence of the infection within the flea and the ability of fleas to pass rickettsiae vertically, readily available host animals may be less important as ongoing reservoirs of infection and more significant as sources of bloodmeals and mechanical carriage of infected fleas to areas near human habitation.

## 5. The Epidemiology and Ecology of Flea-Borne Typhus is Deduced

Epidemic typhus, due to *R. prowasekii*, was recognized as a clinical entity distinct from typhoid after the work of William Gerhard in the United States in 1837 [67]. The principal manifestations of epidemic typhus are fever, headache, rash, altered mental status, peripheral gangrene, myocarditis, and secondary infections; in the pre-antibiotic era, it carried a mortality rate of 20%–60% [68]. In 1909, Nicolle determined that epidemic typhus was vectored by the human body louse, *Pediculus humanus corporis* [69]. In 1910, Nathan Brill of Mount Sinai Hospital (New York, NY) reported a series of febrile patients with a disease that appeared to be mild epidemic typhus [70]. In 1912, James Anderson and Joseph Goldberger of the United States Public Health Service (USPHS) found that in monkeys an attack of Brill’s disease conferred immunity to “Mexican typhus” and that the converse was also true, indicating that Brill’s disease was a mild form of typhus [71]. Initially, the sporadic typhus fever of the Southeastern United States was thought to be Brill’s disease, which was also called endemic typhus [40]. Brill’s disease (now called Brill–Zinsser disease) was determined to be recrudescent epidemic typhus in 1933 [72]. The term “Mexican typhus” was loosely applied to cases of typhus-like illness acquired in or near Mexico and these cases were FBT, epidemic typhus, or Brill’s disease [28,73,74]. The situation was further confused because *R. typhi* may also be transmitted by the human body louse [75].

In 1913, Dr. James Paullin of Grady Hospital in Atlanta, GA was the first to report FBT in the United States; he diagnosed seven cases, based on the fever curves, rashes, negative blood cultures, and negative serologic tests for typhoid (Figure 1) [76]. Shortly thereafter, cases were recognized in Savannah and Augusta, GA, and later in Montgomery and Mobile, AL, and Jacksonville, FL [77]. Four cases were reported from Charlotte, NC, in 1914 by Leon Newell and William Allan [78]; the diagnoses were based on the patients’ prior histories of typhoid or measles, fever curves, negative cultures, and the absence of abdominal symptoms and splenomegaly. In 1916, Dr. H. L. McNeil reported five cases from Galveston, TX [79] and physician Horace Hall of the Texas State Quarantine Service reported a mild typhus-like illness (“Rio Grande Fever”) that occurred in Mexico and along the Texas–Mexico border [80]. In 1918, Dr. John Hunter reported three cases from McAllen, TX, near the Mexican border [13]. Endemic typhus became a reportable disease in the United States in 1920 [81]. In 1924, 70 cases were reported in TX, from Laredo to Weslaco, with Mission having 25 cases [13].

Meanwhile, the first serologic test for a rickettsial infection was developed in Europe in 1915, the Weil–Felix test (WFt), which distinguished typhus (either epidemic or endemic) from other febrile illnesses [84]. In 1917, Mather Neill of the USPHS provided the first experimental evidence for the existence of more than one form of typhus. Neill found that male guinea pigs inoculated with blood from patients in TX infected with “Mexican typhus” (i.e., FBT in this case) showed scrotal swelling and testicular adhesions. This reaction was less severe than that observed when using blood from a Rocky Mountain spotted fever victim but was absent using blood from cases of epidemic typhus and true Brill’s disease (i.e., Brill–Zinsser disease) [85]. In 1928 Hermann Mooser found that rickettsial multiplication occurred in the tunica vaginalis of the testes [86]; this scrotal reaction subsequently became known as the Neill–Mooser reaction. In an era before the availability of specific serological tests and genetic sequencing, the Neill–Mooser reaction provided a method to differentiate FBT from epidemic typhus and Brill–Zinsser disease [87].

In 1919, Frank Hone, a quarantine officer in Australia, observed a number of patients with a mild typhus-like illness; many were employed in grain storage facilities, potentially bringing them into contact with rodents. Hone suggested an insect vector, but he did not make an association with rodents or fleas [88].

In the 1920s to 1930s, several studies further differentiated epidemic typhus, Brill–Zinsser disease, and FBT and determined the epidemiology of FBT in the United States. The most significant work in this regard was carried out by investigators from the USPHS, including Kenneth Maxcy, George Brigham, and the team of R. Eugene Dyer, Aldolph Rumreich, and Lucius Badger (Figure 1 and Figure 2). In 1922, Maxcy, working for USPHS, was assigned to the Alabama State Health Department as an epidemiologist [89]; during investigations of malaria in the American South he became interested in the mild sporadic typhus cases that were being reported there [90].

In 1923, Maxcy and Leon Havens (director of the Alabama State Laboratories) reported 11 patients in AL with fever and a positive WFt. The cases occurred during the summer, fall, and early winter. Ten of the patients lacked louse infestation. No contact with typhus was evident. Maxcy and Havens surmised on clinical and serologic grounds that their cases were most likely Brill’s disease, but they were perplexed by the same epidemiologic characteristics that had befuddled Brill some thirteen years earlier—the lack of communicability and a seasonal distribution different from epidemic typhus [91]. Based on this work, Maxcy was directed by US Surgeon General Hugh Cummings to extend his investigations to other parts of Alabama and neighboring states [92].

In 1925, Maxcy collaborated with Charles Sinclair of the US Army Medical Corps on a study of a typhus-like illness (which they mistakenly called Brill’s disease) in the Rio Grande Valley of TX [93]. From May to June of 1924, 20 patients from several border towns presented a febrile illness of two weeks duration associated with rash. Thirteen of the 20 patients were moderately ill and three were described as critically ill, but there were no fatalities. Unlike epidemic typhus, in these cases neurologic manifestations were not significant. The WFt was positive in 12 of 15 patients. In no case was body louse infestation present, although in 12 of the 20 cases, head lice were observed in the patients or their families. Sinclair and Maxcy also interviewed physicians in several Mexican border states and these physicians reported frequent cases of a similar illness during the summer. The majority of these Mexican patients were also free of body lice. The investigators concluded that the illness was "Brill’s disease", an attenuated form of epidemic typhus, and that the nearly ubiquitous head louse was the most likely vector [93]. However, Drs. Arthur Flickwir (health officer of Houston, TX) and Carl Lovelace (Waco, TX) disputed this because many of the patients "were refined people of good social standing," so obviously a non-louse vector must be involved [13]. In 1926, Drs. Fletcher Gardner and Aubrey Brown of the US Army Medical Corps also described cases from this epidemic, which involved personnel at Fort Ringgold in Rio Grande City, TX. Lice were not present on the victims, but fleas were common, and the authors were the first to speculate that fleas might be the vectors of FBT. Mexican physicians interviewed by the investigators referred to the infection as “Texas Spotted Fever” or “Fourteen-day fever.” Gardner also proposed that endemic typhus may have caused an outbreak of febrile illness involving 20–30 persons at Camp Llano Grande, TX in 1916, which at the time was ascribed to dengue [94].

In 1925, Maxcy transferred to the USPHS Hygienic Laboratory to continue investigations of endemic typhus there. By 1926, Maxcy had assembled the data from his years of fieldwork and he made some crucial epidemiologic observations that would help to differentiate FBT from epidemic typhus [95] and that would ultimately lead to definitive proof regarding the reservoir and vector of FBT. Maxcy noted the seasonality of epidemic typhus in Europe differed from that of the endemic typhus of the Southeastern United States, with the former more prevalent during January through May and the latter June through November. Maxcy also constructed a map to analyze the distribution of cases by place of residence in Montgomery, AL, but found no distinct localization. On the other hand, when he analyzed the cases with respect to the place of employment, he saw a higher attack rate in persons working in groceries, feed stores, and restaurants. From these observations of seasonality, absence of household transmission, and transmission within food establishments, Maxcy proposed a rodent reservoir and transmission by fleas, mites, or possibly ticks, but not by lice [96]. The conclusions of Maxcy are considered to be classic examples of epidemiologic inference [97].

In 1926, Maxcy published the first comprehensive clinical description of “endemic typhus,” based on the records of 114 patients diagnosed in AL and GA from 1924 to 1926. With this data set, Maxcy was the first to quantify the frequency of the signs and symptoms of endemic typhus and provide a natural history of the illness and a thorough description of the rash. Unlike prior investigators, who had diagnosed endemic typhus strictly on clinical grounds, Maxcy’s diagnoses were more accurate due to the availability of the WFt and the Neill reaction. Maxcy correctly surmised that the available laboratory evidence (i.e., WFt positivity) “testifies to the identity or very close relationship of the etiological virus [of FBT] with that of Old World typhus.” [96].

Maxcy reported that African Americans rarely contracted the infection; for example, in Savannah, GA in 1920, only 2.2% of the cases of FBT were observed in African-Americans, even though they comprised 47% of the population [96]. A similar racial disparity was also noted in series from AL in 1935 and 1943 [31,98]. However, it was not known if this was due to true decreased biological susceptibility of African-Americans, lesser access to medical care, underreporting of disease in this group overall, or misdiagnosis due to difficulty in detecting the rash on darker skin [14,31,99]. In a series of 1029 AL cases from 1932 to 1933, the mortality rate for African Americans was 11.7% vs. 3.8% for Caucasians. The most common cause of death associated with FBT, based on death certificates, was pneumonia, followed by nephritis, myocarditis, and apoplexy (i.e., altered mental status). The higher death rate in African Americans may have been due to incomplete registration of milder cases or more severe cases occurring in the setting of glucose-6-phosphate deficiency, which is more common in African Americans [100,101,102]. The death rate was also higher in older persons, likely because of pre-existing conditions in the heart, lungs, or kidneys [100].

Maxcy published a review of his work on FBT in 1929; by that time, he had identified cases along the lower Atlantic seaboard, in multiple Gulf Coast ports, and in the lower Rio Grande Valley, TX [92]. No cases had yet been reported from MS, LA, and TN. Later in 1929, Maxcy left the USPHS and FBT investigations at the USPHS were assumed by Dyer, Badger, and Rumreich, who were later joined by William Workman and Elmer Ceder [89,103]. A breakthrough in the epidemiology of FBT was achieved in 1930, when a Baltimore physician-pharmacist fell ill with a fever, headache, and rash. The WFt was performed and the titer was high at 1:320. Two other workers in the same pharmacy were also stricken with a similar illness over the next three weeks. The USPHS dispatched Rumreich to the scene, whereupon he trapped rats that were living in the pharmacy basement. Using suspensions of the fleas (*X. cheopis* and *Ceratophyllus* (now *Nosopsyllus*) *fasciatus*) obtained from these rats, Dyer, Rumreich, and Badger then successfully transmitted the pathogen of FBT to guinea pigs. Thus, they had identified both the vector and the reservoir of FBT [40,104]. The FBT organism was also recovered from fleas (*X. cheopis* and *L. segnis*) taken from rats trapped at an FBT focus in Savannah, GA [105]. Prior to this work in the early 1930s, the specific arthropod vectors of FBT were still unknown; the possibilities included chiggers, mosquitos, lice, bedbugs, ticks, fleas, and mites [106].

The role of rats (*Ra. rattus*) as reservoirs was also demonstrated in 1931 by Hermann Mooser and Maximiliano Ruiz Casteñada of the American Hospital in Mexico City and Hans Zinsser of Harvard Medical School (Figure 3). During an outbreak of “Mexican typhus” in Mexico City they produced FBT in guinea pigs using brain emulsions of rats that had been trapped in areas of the outbreak and demonstrated the presence of rickettsiae in the guinea pigs [107]. The reservoir role of rats was consolidated when Dyer, Workman, and Rumreich recovered rickettsiae from rats trapped at an FBT focus in GA in 1932 [108]. In 1931, Dr. Hardy Kemp showed that *X. cheopis* and *N. fasciatus* taken from wild rats, pulverized, and injected into guinea pigs produced a typhus-like illness, thereby further implicating these fleas as vectors [109]. Soon thereafter, Dyer and coworkers showed that *N. fasciatus* could transmit FBT by natural means [110]. Rumreich, Dyer, and Badger differentiated FBT from Rocky Mountain spotted fever (RMSF) on the basis of clinical and epidemiologic characteristics in 1931 [16,111].

In 1931, Dyer, Ceder, Rumreich, and Badger established that *X. cheopis* can harbor *R. typhi* and transmit the infection from rat to rat. The investigators introduced hungry fleas into a cage containing *R. typhi*-infected rats and uninfected rats, which led to infection of the latter [112]. In that same year, Mooser, Ruiz Casteñada, and Zinsser found that the spiny rat louse *Polyplax spinulosa* was the most abundant ectoparasite on rats in Mexico. They also determined that there is tremendous replication of rickettsiae within these lice and that they are able to transmit rickettsiae from rat to rat. However, this louse is not an important vector to humans [113]. Later in 1931, Walter Dove, an entomologist at the US Bureau of Entomology and Plant Quarantine, and J. Bedford Shelmire Jr., a Baylor University dermatologist, determined that the tropical rat mite *Ornithonyssus bacoti* was able to transmit FBT between guinea pigs and they implicated this mite as a potential intermurid vector and a possible vector to humans due to its propensity to attack humans [114,115,116]. Subsequently, another investigators confirmed its vector potential [117] (rats may harbor multiple ectoparasite species simultaneously; Strandtman and Eben reported a rat that was infested by eight different arthropod species: three fleas, four mites, and one louse [116]). However, studies by William Smith of the USPHS in the 1950s did not support the significance of either the tropical rat mite nor the spiny rat louse in the transmission of *R. typhi* among rats [118].

By the end of 1931, Ceder, Dyer, Rumreich, and Badger demonstrated that feces from infected fleas were capable of transmitting FBT through scratches on the skin of guinea pigs, indicating that a major route of transmission is through inoculation of infected feces into abraded skin or mucous membranes [50,119] (subsequently, it was shown that a flea bite may also transmit the infection [2,39]). In the following year, the same team discovered that the feces of *X. cheopis* become highly infectious six days after feeding on *R. typhi*-infected rats. An inoculum as low as 1/128,000 of a flea produced infection and the fleas remained infectious for at least 42 days [50]. The importance of *X. cheopis* as a vector of FBT was further solidified when it was determined that rats infested with this flea were 70% more likely to be seropositive for FBT compared to rats infested with other flea species [31]. In 1933, Brill’s disease was determined to be a recrudescent form of epidemic typhus by Zinsser and Ruiz Casteñada [72,120]; it was renamed Brill–Zinsser disease in 1952 [121].

In 1932, Mooser and Ruiz Casteñada defined the events transpiring within the flea after the ingestion of rickettsiae. They found that the rickettsiae multiplied within the epithelial cells of the stomach but were prevented from entering the lumen of the gut in quantity by the peritrophic membrane which lines of the gut. However, rickettsial multiplication was also found to occur within the Malpighian tubules (excretory organs), and this was the likely source of the bacteria in the flea feces [122].

By 1938, Brigham and Dyer established that other native rodents were susceptible to FBT, including: mice in the genus *Peromyscus*, wood rat *Neotoma floridana,* cotton rat *Sigmodon hispidus*, rice rat *Oryzomys palustris,* and flying squirrel *Glaucomys volans* [123,124]. Although these rodents may not be directly consequential to human FBT, they help maintain the causative organisms in nature. Subsequently, rabbits, other squirrels, woodchucks, cats, dogs, and skunks were found to be susceptible, whereas raccoons and foxes were resistant [123,125]. In 1941, Brigham isolated FBT rickettsiae from chicken fleas obtained from rats on a GA farm with a farmer ill with FBT [126]. The cat flea was first implicated as a vector in 1942 when Jessie Irons and colleagues of the Texas State Department of Health recovered *R. typhi* from naturally-infected cat fleas. The discovery was made when four members of a family contracted FBT after they had acquired a kitten. The index kitten was no longer available, but its littermates bore many infected fleas [127].

## 6. The Increasing Incidence of Flea-Borne Typhus and Initial Efforts at Control: 1930–1944

In the 1920s and early 1930s, the number of reported cases of FBT in the American South was increasing almost year after year. In 1922, 48 cases were reported; each successive year showed an increase until 239 cases were reported in 1929, although data obtained before 1929 are considered less accurate [128]. For 1930 through 1933, there were 511, 333, 957, and 2070 reported cases, respectively [81]. During the 1920s, Maxcy observed FBT in port cities of the southern Atlantic seaboard and the Gulf of Mexico and only an “occasional case from the interior of the country, that section has been for the most part strikingly free” [92]. However, by 1930, FBT was moving inland and from cities into rural areas [124]. Rats are notorious stowaways and rat migration (including their fleas) was facilitated by the burgeoning rail and road networks of the American South in the 1920s [56,128]; the number of paved roads in the South increased significantly after the initial impetus supplied by the Federal Highway Act of 1921, from 121,164 miles in 1921 to 209,880 by 1930 [129]. The displacement of the black rat by the brown rat in many areas of the South during the 1930s and 1940s may have also contributed to the increase in FBT cases [56,128] because the larger brown rat is more likely infested by *X. cheopis*, maintains a higher flea burden than the black rat, and is more likely infected by FBT [56,59,118]. Whether this differential in infection rates is due to some greater biologic susceptibility to FBT infection in the brown rat or a consequence of the higher flea burden is unknown.

By 1930, the WFt was accepted as a specific test for typhus by the USPHS and adopted by state health departments; this led to greater case ascertainment throughout the 1930s [128]. By the early 1930s, due to the investigations of Dyer and coworkers, the links of FBT to its rodent reservoirs and flea vector was established. With an untreatable disease of considerable morbidity, increasing in incidence, with known epidemiology, in the early 1930s, the USPHS instituted rodent control measures, which consisted of rat proofing buildings and rat extermination [130]. In 1933, President Franklin Roosevelt authorized the creation of the Civil Works Administration (CWA) [131]. One of the duties assigned to CWA workers was rat control. From 15 December, 1933 to 29 March, 1934, some 10,000 CWA workers baited 747,608 premises in GA, AL, and TX with the rodenticide red squill and exterminated over 7.5 million rats [132]. In TX, one million rats were killed in 1933 [13] (Figure 4). Four million rats were eradicated by the CWA in 21 of Alabama’s 67 counties (Figure 4) and typhus cases in 1934 decreased by 79% compared to 1933 [98]. However, these rat control programs were discontinued in 1934. Nevertheless, the drop in the number of cases of FBT in the United States during 1934–1936 is attributed to these control programs [128,133]. However, another factor that likely contributed to a decrease in cases in 1934 and the years immediately following was that TX experienced a severe drought in 1934 that was accompanied by a rat die-off [134]. In 1936, the Works Progress Administration instituted a rat trapping project to test rodents for the presence of various infections (Figure 5, Figure 6 and Figure 7) [135]. The most highly endemic areas of the country for FBT in the 1920s and 1930s were southern GA, southeastern AL, and southeastern TX [128]. In AL, the incidence of disease increased from about 80 cases/year in 1931 to 237 in 1932 and 823 in 1933 [98]. In the period 1922–1939, GA reported 6225 cases of FBT; AL, 3751 cases; and TX, 3277 cases, with 84%, 85%, and 62% of the counties in these states, respectively, reporting cases. Savannah, GA, was the city affected worst, with 1167 cases from 1921 to 1939. Two other high incidence cities were Atlanta, GA and Charleston, SC, with 354 and 309 cases, respectively, during the same period [128].

In the pre-antibiotic era, in two hospital-based series (Grady Hospital in Atlanta, GA and Charity Hospital in New Orleans, LA), the mortality rate of FBT was less than 1% [137,138]; however, mortality rates ranged from 3.6% to 7.6% in published series from 1922–1944 [5,16,98], and were significantly higher (30%) for those older than 65 years [100]. The Baltimore City Health Department reported a 22% mortality rate in 59 patients from 1927 to 1947 [139]. For those patients that succumbed, death typically occurred in the second week of illness [92]. Although the mortality rates of FBT were not high, the disease did cause significant morbidity and economic hardship. In 1933, 43% of the 53 cases reported from Atlanta required hospitalization (although undoubtedly many mild cases escaped recognition) [16]. In the pre-antibiotic era, these hospitalizations typically lasted two weeks and a 4- to 8-week period of convalescence was necessary [140,141,142,143]; this could be financially devastating to a family in an era of one breadwinner and the absence of medical insurance and social safety nets. In 1940, it was estimated that the typical cost for medical care and lost productivity for an FBT patient was $154 [142], compared to $389 for the average annual per capita income in the American South at that time [144].

Due to the increasing incidence of FBT in GA, from 51 cases and 1 death in 1929, to 1092 cases and 54 deaths in 1937, the Georgia Department of Public Health established a Typhus Control Unit in 1937. Its goals were to: (1) formulate an action plan for statewide control; (2) provide expertise to county and municipal governments; and (3) organize local rat control programs [56]. Due to concern about the increasing number of cases of FBT, the USPHS and Georgia Department of Public Health held a training course on rat control in Macon in 1937. At this meeting, a cadre of public health officials of the Southeastern region that were later to be active in FBT control received their initial training [133].

Initially, the focus of rat control programs was “complete rat proofing” (eliminating all dead spaces between walls and flooring) [145], but this was found to be too expensive for older buildings, so the emphasis was changed to “vent stoppage,” (i.e., closing of openings in the exterior walls of buildings with cement or metal impervious to gnawing) [56,146]. The Georgia Department of Public Health started typhus control projects in 1937 which focused on improved sanitation and rat control. In GA at that time, the most abundant rat was the brown rat and the most common fleas were *N. fasciatus* and *X. cheopis*. Urban areas in southwest GA had the highest incidence of FBT in the state. Most of the urban cases occurred in persons associated with food handling establishments (restaurants, grocery stores, markets, and warehouses). Before a control project was undertaken, it was necessary to assess the foci of infection and various attributes of the area (prevalence and species of rats; types and conditions of buildings and their patterns of use (i.e., food-handling vs. other); harborage sites; garbage disposal conditions; and the economic status of the inhabitants. Community education was deemed important to the success of the program, and all available media of that era were employed: newspaper articles, radio talks, demonstrations, exhibits, leaflets included in utility bills, and lectures with slides given to schools and civic groups [56].

In over two years (1937–1939), the Georgia Typhus Control Unit placed 14,470 lbs of rat bait laced with red squill. Poisoning was found to be more effective if garbage removal and harborage elimination were initiated first. Trapping was also performed, but it was costly compared to poisoning. Commercial buildings were surveyed, and an assessment of the cost of rat proofing was given to each owner; the average cost of rat proofing was about $25 per structure. The most expedient method of rat proofing was vent stoppage. The rats remaining in the buildings were then poisoned or trapped. Nevertheless, Roy Boston of the Georgia Department of Public Health remarked that typhus control was often deemed important in a community only after an outbreak had occurred, especially if some of the leading citizens were affected [56]. Alabama implemented a similar program in 1938 [133]; efforts were made in various cities to remediate buildings for greater resilience to rat intrusion [147].

At a meeting of the American Medical Association in 1937, Dr. John J. Phair declared that the elimination of rat populations in rural areas was practically impossible. Furthermore, intensive use of rodenticides outside of the city would also threaten local wildlife. Phair proposed that a more realistic goal in rural areas would be to reduce rat populations to a level at which transmission of FBT would be uncommon [148].

In 1937, the USPHS published a comprehensive guide to rat proof construction, replete with photographs and detailed diagrams prepared by architect Philip Clark [149]. Many of these construction specifications were based on studies conducted by the USPHS that involved captive wild rats to determine to what depth they could burrow, their swimming capabilities, and their ability to climb various types of pipes and exterior walls [150].

By 1939, the USPHS assembled a team of specialists under the direction of Assistant Surgeon General Dr. Charles Williams to act as advisors on rat control projects [133]. An example of the utility of this program was the management of an outbreak of 75 cases of FBT in Nashville, TN in 1939. The main focus of FBT in Nashville was near granaries close to railroad tracks entering the city. Thus, this epidemic illustrated how FBT spread via infected rats arriving on rail cars. After the implementation of rat proofing, the extermination of 14,000 rats, and improvements in urban sanitation, as assisted by the USPHS, the number of cases of FBT in Nashville dropped to 18 in the following years [99].

In the 1940s, to promote rat extermination, some communities in TX placed a five-cent bounty on rattails [151]. In May 1942, the Typhus Control Unit (TCU) of the USPHS was established, which provided technical expertise and financial assistance to local rodent control programs. An emphasis was placed on exterior rat proofing in military and war industry zones and environmental hygiene (Figure 8 and Figure 9). Local authorities were expected to: (1) enact suitable sanitary ordinances; (2) provide trainees so that the program could continue after the USPHS withdrew its own personnel; (3) provide laborers for construction and rat trapping; and (4) obtain funding to purchase materials to execute the project. The TCU furnished sanitary inspectors capable of conducting various activities of the project, which included: (a) assessing the extent of rat infestation of each building, the measures necessary for rat proofing, and the cost of each job; (b) to supervise the rat proofing, trapping, and poisoning operations; (c) to train local personnel; (d) to provide personnel and apparatus for cyanide fumigation; and (e) to lend rat traps. The TCU typically provided support for four to six months after which local agencies would assume continuation of the project. Once positive results were obtained in a community, “word of its value in reducing rat damage soon spreads from one businessman to another and makes extension of the project relatively easy” [152]. From May 1942 to June 1945, the TCU participated in 61 rodent control projects [133]. Despite these measures, the number of cases in GA rose to 1135 in 1939 and remained high at 1111 in 1945 [153]. In 1945, the incidence rates in southern GA counties ranged from 202 to 218 per 100,000 residents [154].

Dr. Leon Banov, Director of the Charleston County (SC) Health Department described the local efforts to control rats in Charleston prior to World War II in his memoirs. Health department personnel set up an assembly line to construct "torpedoes" consisting of meat, fish, or grain laced with red squill wrapped in paper (Figure 10). The plan was to simultaneously distribute the poisoned baits all across the city on a specific day. A large troupe of civic-minded women was recruited to go door-to-door selling the torpedoes to the residents of Charleston, SC. Efforts were also undertaken to perform vent stoppage on local buildings [155].

Before World War II, pest control was limited by the primitive state of available technologies. In the early 20th century, available insecticides included arsenicals, lime-sulfur, petroleum oils, pyrethrum, *p*-dichlorobenzene, carbon disulfide, naphthalene, and nicotine. In the interwar period, phenothiazine, rotenone, dinitrophenols, and thiocyanates were introduced into insect pest management [157]. Pyrethrum, isolated from *Chrysanthemum* flower heads, is a contact poison with rapid onset of toxicity to insects, but it is unstable in air and light, and has little residual activity [158]. Rotenone, derived from the Asian plant *Derris elliptica*, has a slow onset of action and little residual activity [159]. Nicotine, from tobacco, is expensive to produce, extremely toxic to humans, and susceptible to degradation [160].

Powdered naphthalene was recommended for flea control, but it requires airtight conditions for a period of 24 h. Sodium fluoride powder was also used as an insecticide in the 1920s and 1930s, but its efficacy was uncertain [161]. In the first forty years of the 20th century one of the principal means to destroy vermin in buildings or ships was fumigation with hydrogen cyanide gas or one of its precursors, such as cyanogen or cyanogen chloride. Hydrogen cyanide gas, generated in situ by mixing a cyanide salt with an acid, was considered cheap and effective. Unfortunately, it sometimes proved as deadly to people as to pests. In 1933, 74,000 fumigations were performed in the United States, with one human death per 6000 fumigations [162]. Other fumigants used in the 1930s included sulfur dioxide (generated in situ by sulfur combustion) and carbon disulfide [163]. However, sulfur dioxide bleaches fabrics, tarnishes metals, and has poor diffusability, necessitating long exposure times. Carbon disulfide is flammable and had uncertain efficacy [161,163,164]. Calcium cyanide was also used; this material generates hydrogen cyanide in situ on atmospheric contact (Figure 11). It is easier to handle than a gas but requires specific conditions of temperature and humidity to work effectively [161]. On a single application to rat burrows, its efficacy was only about 50%. Furthermore, care was necessary to avoid inhaling the powder and for complete ventilation of dwellings prior to human re-occupation. Furthermore, fumigation had inherent limitations: dwellings had to be airtight for the treatment to be effective and fumigants afforded no protection against re-infestation [158]. The primitive state of flea control in this era is clearly illustrated by advice provided in a textbook of parasitology from 1936: “Various traps for the capture of adult fleas have been devised, one of the simplest and most effective being to clothe the legs in sticky fly paper and wander about the infested rooms. A badly infested building in Cornell University was cleared of fleas in this manner.” [163].

Furthermore, the supply of some of these early insecticides became limited by 1942. Acquiring reliable supplies of pyrethrum during wartime had become problematic. In 1939, the United States was importing 13.5 million lbs of pyrethrum from Japan and Kenya, with 90% from the former. With the advent of war, the Japanese supply was abruptly terminated [165] and all supplies of pyrethrum were placed under control of the US War Production Board. A drought in Africa in 1942–1944 decimated the *Chrysanthemum* crop. The military also increased its use of pyrethrum for the control of mosquitos and lice [166]. Thus, by 1943, there was a global shortage of pyrethrum [167]. Domestic supplies of rotenone were also depleted by military use and the loss of *Derris*-growing areas by the fall of the Dutch East Indies and British Malaya to the Japanese. Another demand imposed on the American insecticide supply during World War II was the Victory Garden movement, which started in late 1942. By 1944, there were 22 million home gardens, a 180% increase over the usual number of gardens. To maintain productivity, these gardens required 25,000 tons of insecticide per year, further depleting the available pesticide stocks [166].

Rodenticides used in the early 20th century included arsenic, strychnine, and phosphorus [168,169]. Arsenic oxide was employed as a rodenticide since the 17th century [170]; it was moderately effective, but variable in quality [171]. The application of strychnine as a rat poison dates from the same period, but its utility was limited by its poor palatability [172]. Barium carbonate was used as a rodenticide since the late 19th century, but it also suffered from low palatability. Phosphorus was first utilized as a rodenticide in the mid-1800s, and its derivative, zinc phosphide, entered service in this role the late 1930s. Thallium sulfate was introduced into rodent control practice in the 1920s [168], but its extreme toxicity was a potential hazard to non-target species [145].

Red squill, a rodenticide derived from the bulb of *Urginea maritima*, was first employed in 1781 [168] and achieved widespread use in the United States in the mid-1920s because it was less toxic to other mammals than the alternatives [169]. However, batches of red squill varied in potency [158] and deteriorated upon exposure to moisture or light [168]. Another deficiency of red squill that compromised its efficacy was that ingestion in sublethal amounts caused aversion in rodents [169]. Overall, the poor palatability of the available rodenticides was an obstacle limiting the success of the rat control programs of the 1930s and early 1940s. Clifford Eskey, USPHS medical director, lamented that with the rodenticides available in the early 1940s, “In the southern region [FBT] seems to be so widely established among the rat population that it can never be completely eradicated.” [146] Despite more than a decade of rat proofing, trapping, and extermination, the unfortunate conclusion is that these rodent control strategies failed to curb the incidence of typhus in the United States [173]. From the time that CWA rat control activities commenced in 1933 to the apex of the epidemiologic curve in 1944 (Figure 12), the annual number of reported cases of FBT increased from 2070 to 5401 [81].

During World War II, imports of various rodenticides to the United States were terminated [174,175]. Germany was the primary source of thallium prior to World War II [176]. Strychnine, isolated from the Asian tree *Strychnos nux-vomica*, was also in short supply due to Japanese military activity [174,175]. Red squill, imported from the Mediterranean basin, also became difficult to obtain [177]. As a result of these shortages of traditional rodenticides, the use of zinc phosphide became more widespread during World War II, but it was not especially effective and highly toxic to humans and domestic animals, limiting its use [168,178]. Effective and economical rat proofing was often limited by the dilapidated condition of the buildings of that era [31,179]. Furthermore, rat poisoning without the concurrent use of an effective insecticide has the problem in FBT control in that as the rats die, the fleas will abandon the carcass and will seek other hosts, including domestic animals and humans [134,158].

Flea-borne typhus cases increased throughout the 1930s and into the 1940s. In 1930, 511 cases were diagnosed; 2070 in 1933; 2393 in 1937, 2996 in 1939, and 5401 in 1944 [180]. Overall, there were 42,000 reported cases of FBT in the United States between 1931–1946 [1], with 95% of the cases reported from eight states: NC, GA, SC, FL, AL, MS, LA, and TX [181]; 90% occurred south of a line drawn from Charleston, SC, to Dallas, TX [81]. The heaviest focus laid in a belt extending from coastal SC and GA westward along the Gulf of Mexico into central TX [181]. A progressively greater incidence of FBT was reported more southerly within this zone [133]. Above 33° north latitude (roughly the border between LA and AK), most cases occurred in workers of food establishments in business districts of urban centers [133,146,182]. Flea-borne typhus was most prevalent in areas where the average January temperature was above 4 °C and the average relative humidity in July at noon was above 37% [183]; these conditions are favorable for rat flea proliferation [34]. Seasonally, the lowest number of FBT cases was reported in the fall and winter and the greatest in July through September. With the exception of 1940, the number of cases reported increased each year since surveillance of the disease started in the early 1920s; the decrease in cases in 1940 was likely due to an unusually cold January that year. There was no association between precipitation and the incidence of FBT [181]. Although by 1946, typhus was reported in 37 states and the District of Columbia, 67% of cases occurred in just 100 counties in nine Southern states (NC, SC, GA, AL, FL, MS, LA, TX, and TN); the highest rates per county were 430 per 100,000 population. In the highest incidence areas (southwest GA, southeast AL, multiple TX foci), about half of the cases were contracted in rural and residential settings [182]. Furthermore, despite these high numbers, the actual incidence of FBT was thought to be much higher than the reported incidence because many cases were not brought to medical attention or because it was mistaken for other infections [31,146]. In a series of 56 patients with FBT in CA from 1924–1946, FBT was initially suspected in only two of the cases; other initial impressions included pneumonia, typhoid, influenza, meningitis, the common cold, brucellosis, measles, septicemia, mononucleosis, tuberculosis, encephalitis, leptospirosis, smallpox, and plague [184]. For example, in Coffee County, AL, 61 cases of FBT were reported in 1943. However, door-to-door surveys suggested that 211 persons may have had the disease, and 135 of 177 collected human sera specimens gave positive complement fixation or Weil–Felix tests [31]. In San Antonio, TX, Davis and Pollard performed the complement fixation serologic test on 4219 persons in 1945 and estimated that probably 700 persons were infected each year between 1935 and 1945, even though the maximum number of cases reported to the San Antonio Health Department was 91 in 1944 [185]. Based on a survey conducted in four counties in southwestern GA, only one-third of the cases that occurred in 1945 were reported to health authorities [173]. In FL, a serosurvey of FBT conducted in 1944–1946 concluded that cases in children and African Americans were under recognized [186]. Overall, it was thought that the number of cases of FBT nationwide was underestimated by 3- to 5-fold [146,180,182].

## 7. Factors in the Rise of Flea-Borne Typhus in the United States 1913 to 1944

The factors that caused the dramatic rise in the number of reported cases of FBT in the United States in the period 1935–1945 are not entirely clear. Certainly, there was increased recognition of the disease due to the work of the USPHS and the availability of the WFt [187]. Sherman and Langmuir suggested that the slower increase from 1932 to 1940 was due to greater recognition of the disease and the upsurge from 1940–1944 was a true increase in incidence [188]. The increase in cases after 1930 was not primarily due to an extension of the range of the disease, but rather an increase in cases within the established core endemic area. However, the large increase in the number of cases in the period 1935–1945 was likely due to more than just improved case ascertainment. The increasing incidence of FBT must be examined within the context of social conditions that prevailed in the American South in the 1930s and 1940s.

The economic and social milieu of the American South from the 1920s, through the Great Depression, and into the World War II years, provided an ideal incubator for FBT. The years 1917–1919 brought record high prices for cotton. However, after the brief prosperity of the World War I era, throughout the 1920s, the South was beset by a series of natural and man-made calamities. In 1920, the most dramatic price collapse in cotton history occurred. Although cotton prices would rise again, the agricultural crisis of 1920 dashed the hopes of Southern farmers to attain a modicum of financial security [189].

In 1921, the boll weevil destroyed 30% of the cotton crop and spread throughout the Cotton Belt by 1923 [190]. As the boll weevil marched eastward, it left in its wake both economic devastation and an outcry for crop diversification [129]. In 1927, a great Mississippi River flood brought famine to parts of the South. In 1928, numerous Southern banks failed from a persistent absence of farm profits for the last nine years [190]. Two-thirds of the poorest farms in the country in 1929 were in the South [191]. Thus, even before the Great Depression, the South was a land of utter destitution [190].

Then the Depression hit. By 1930, on 56% of all Southern farms the labor was provided by tenants. By 1932, incomes had declined to 58% of their 1929 levels; farm incomes would later drop to 39% of that of 1929. Bank failures continued in the South in 1929 and 1930 at a greater rate than in the rest of the country. Only three years after the great flood of the Mississippi, a record-setting drought parched the South. Annual incomes in the South dropped from $372 in 1929 to $203 in 1932, vs. $797 to $448 outside the region [129]. The value of residential construction fell to all time low in 1933 [192]. 

Cotton prices plummeted to crisis levels again in 1932. In 1933, another bumper crop of cotton, with attendant low prices, was predicted. In 1934, the Bankhead Cotton Control Act was passed by Congress to limit cotton production [193]. Furthermore, the demand for Southern cotton had plummeted on the world market, from 60% of all cotton used abroad to only 23% in 1937. Competition from paper, rayon, and other synthetics further crippled King Cotton. As a result of governmental action and market forces, cotton acreage in the South declined from 44,768,000 in 1929 to 22,800,000 in 1939. Additionally, decades of monocrop cotton cultivation depleted Southern soils. In 1935, the Soil Conservation Act was passed, and new crops, such as soybeans, sorghum, and hay were adopted to protect and replete the soil [129].

In place of cotton, other feed and food crops gained significantly. From 1929–1946 there were increased yields of corn, peanuts, soybeans, truck crops, rice, sugar, citrus fruits, and peaches on Southern farms. By 1939, the Southeast led all other regions in proportions of farms with hogs and poultry [129]. World War II increased demand for oil-producing crops and the acreage planted in peanuts increased 79% from 1941 to 1942 (1.9 vs. 3.4 million acres) [194]. Thus, the net effect of limitations on cotton production, decreased demand for cotton, and the ravages of the boll weevil was the adoption of food crops and livestock in place of cotton. World War II further accelerated this trend. The new food crops and availability of livestock and poultry feed provided more food for rats [179]. Peanut production in the South increased from 800 million lbs/yr in 1930 to 1800 million lbs/yr by 1940 [195]. Peanut hay, the residue left after peanut harvesting that was saved to feed livestock, provided both food and harborage for rodents [42]. In areas were cotton was abandoned on a large scale and peanuts were adopted in its place, such as southern GA and AL [129], FBT had the heaviest impact. It was no coincidence that the states with the largest peanut crop (AL, GA, and TX) [194] also had the most cases of FBT [128]. In GA and AL, FBT cases were clustered in the regions of greatest peanut production, the southwestern and southeastern corners of the states, respectively [77,98,100]. For example, in 1943, the incidence of FBT in Coffee County in southeastern AL was 500 cases/100,000 [31] (by comparison, the incidence rate of Lyme disease in CT, the state with the highest incidence, was 98/100,000 in 1999) [196]. At that time, 42% of rats trapped in Coffee County had a positive complement fixation test for FBT [31]. By contrast, in states that remained shackled to cotton production, such as MI and LA, FBT had a lesser impact [100,128] (Table 2).

In addition to the transformation of Southern agriculture, the generally poor condition of housing in the South in the 1930 and 1940s also facilitated rodent proliferation. By 1935, one of every four Southerners lived in a tenant farmer family. The tenant farmers occupied the worst housing in the nation [197]; home was typically a dilapidated cabin, which was without a ceiling, unscreened, lacking electricity and running water, and covered with a leaky roof [131,198]. Historian Thomas Clark, who witnessed the conditions of the South during the Depression, stated: “In 1935 vast areas of the rural South were reduced to shabbiness. Farmsteads were cluttered and run-down rusting implements and vehicles were scattered about in the disarray of abandonment.” [199]. Historian Clifford Roland described the South of the World War II-era as “an archaic rural area. Compared with the tidy communities and farms of New England, the great barn country of Pennsylvania, or the geometrically sectioned face of the Midwest, the rural south was irregular and unkempt, a bushy, weedy land of unpainted farmhouses, dog run cabins, sharecroppers’ huts, tumbledown barns, and makeshift sheds.” [200]. Of course, such an array of ramshackle structures and clutter afforded ample harborage for rat and was nearly impossible to render rodent-proof. In addition, to eke every bit of income from the available land, planting was done up to the front door of the tenant shacks [129,197]. This also facilitated domestic rat infestation, because rats went directly from the fields into the houses; otherwise rats avoid crossing open areas [201]. The depressed cotton economy initiated a migration from the countryside to the city, and Southern cities were unable to accommodate the influx. Overcrowding was common, with as many as eight families in one house and five persons per room. The urban housing situation in the South worsened throughout the Great Depression [202].

World War II wrought other changes in the South as well. For strategic reasons and economic development, the federal government specifically targeted the South as a location for military training centers, defense plants, and shipyards. Industrialization due to the war effort caused a migration from rural areas into cities, which were unprepared for the influx of people. The urban population of the South increased 36% in the 1940s, with most of the increase occurring during the war years [203]. Decent housing filled quickly, and all manner of buildings were converted into apartments. There was no regard to urban planning or zoning. Sanitation and trash collection failed [198,203]. Mobile, AL, with two shipyards and an aluminum factory, provided a classic example of wartime boom and the strain it placed on public services. In 1940, the population of Mobile was 114,906. Four years later, its population had ballooned to 201,369 [204]. War workers swarmed into abandoned tenant shacks, tent colonies, and trailer camps. Hotels filled and old buildings became dormitories that rented beds in shifts [129]. In describing Mobile in 1943, author John Dos Passos remarked “Gutters are stacked with litter. Garbage cans are overflowing. Frame houses bulge with men.” [205].

Agnes Meyer, a reporter for the *Washington Post*, vividly described the deplorable housing and sanitary conditions in several Southern cities inundated with war workers in her 1943 book *Journey through Chaos* [206]. In Birmingham, AL, *Time* magazine reported that there were 50 cases of murine typhus in the first ten months of 1944, compared to a pre-war annual average of five cases, a situation due to wartime neglect of rat control and a shortage of garbage cans with lids [207]. Thus, conditions in the urban South during World War II were also ideal for rodent proliferation: substandard housing and poor environmental sanitation [61]. It was not until mid-1944 that these living conditions began to improve due to federally-funded services and housing [129].

During World War II, from 1941–1945, about 20,000 cases of FBT were reported to the USPHS, a five-fold increase in the average annual number of cases as compared to 1930 [187]. The factors that led to this increase included: the poor condition of rural and urban housing; the migration of the larger brown rat into the South; poor environmental sanitation in urban areas; the loss of insecticides and rodenticides that previously originated in military theaters of operation; decreased civilian availability of pesticides due to wartime demands; and the replacement of cotton by food crops on many Southern farms due to economic factors and agricultural policies. Alternatively, rickettsiologist Joseph Smadel proposed that the period of 1935 to 1945 was part of a poorly understood cycle of waxing and waning of rodent populations, as occurs in bubonic plague, and that higher rates of FBT were due to larger cyclical populations of rats. His speculation was based on the occurrence of a global increase in plague that started in the 1930s and dissipated by the 1950s [5]. However, clearly both urban and rural conditions in the American South during this period were ideal for the proliferation of rodents and their ectoparasites.

## 8. Conclusions

Flea-borne typhus, due to *Rickettsia typhi* and *R. felis*, is an infection causing fever, headache, rash, and diverse organ manifestations that can result in critical illness or death. It probably invaded the United States through its Southern and Gulf Coast seaports and across the Texas–Mexico border in the early 20^th^ century. Flea-borne typhus was an emerging infectious disease, primarily in the Southern United States, from 1910 to 1944. In that period, the clinical characteristics of the infection were described, and it was differentiated from epidemic typhus, recrudescent epidemic typhus, and Rocky Mountain spotted fever. The period 1930 to 1944 saw a dramatic increase in the number of reported cases. Part of the rise was due to greater recognition of the disease, but there was also a true increase in the number of cases. The incidence of FBT increased between 1930 and 1944 because of conditions favorable to the proliferation of rodents and their fleas during the Depression and World War II years, including: dilapidated, overcrowded housing; poor environmental sanitation; and the difficulty of importing insecticides and rodenticides during wartime. American involvement in World War II, in the short term, further perpetuated the epidemic of FBT by the increased production of food crops in the South and by promoting crowded and unsanitary conditions in Southern cities due to industrialization and the siting of military encampments.

## Figures and Tables

**Figure 1 tropicalmed-05-00037-f001:**
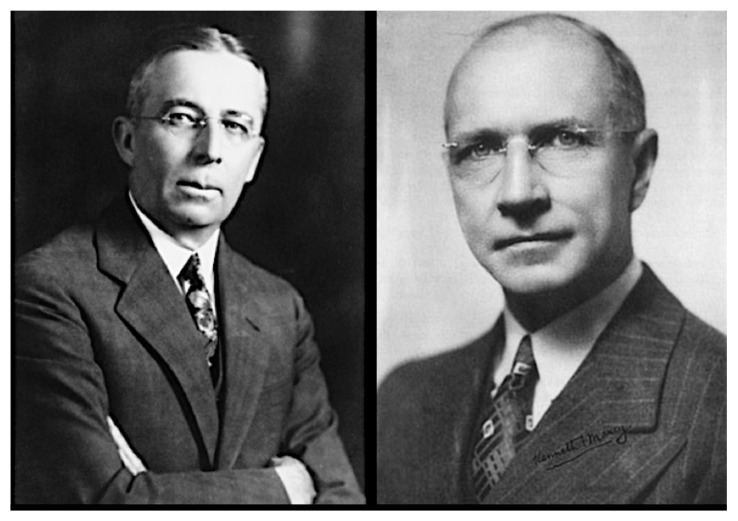
The Pioneers. (**Left**) James E. Paullin. Paullin described the first cases of flea-borne typhus (FBT) in 1913 while at Grady Hospital in Atlanta, GA. Paullin later served as physician to President Franklin Roosevelt [82]. (**Right**) Kenneth F. Maxcy of the US Public Health Service. Maxcy was among the first to propose a rodent reservoir for FBT and published the first comprehensive clinical description of “endemic typhus”. In 1952 Maxcy was awarded the Sedgwick Memorial Medal for Distinguished Service in Public Health, the most prestigious award of the American Public Health Association [83].

**Figure 2 tropicalmed-05-00037-f002:**
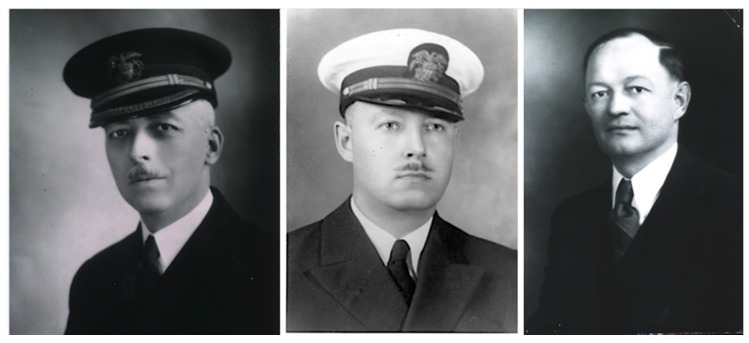
Typhus triumvirate of the United States Public Health Service: (**left)** Eugene Rollo Dyer; (**center**) Adolph Rumreich; (**right**) Lucius Badger. Along with William Workman and Elmer Ceder, these men determined that FBT has a domestic rodent reservoir and is transmitted by the flea feces and discovered the vector potential of the Oriental rat flea. In 1931, the team also differentiated FBT from Rocky Mountain spotted fever (RMSF). For his work on typhus, Dyer was awarded the Sedgwick Memorial Medal for Distinguished Service in Public Health in 1950 [83].

**Figure 3 tropicalmed-05-00037-f003:**
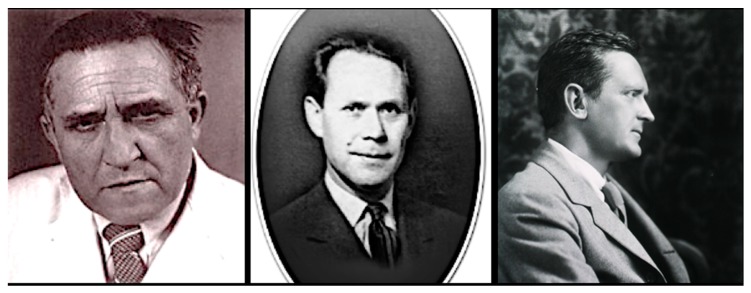
Typhus troika working at the American Hospital in Mexico City: (**left**) Hermann Mooser; (**center**) Maximiliano Ruiz Casteñada; (**right**) Hans Zinsser. They also determined that rats harbor the FBT rickettsiae [107].

**Figure 4 tropicalmed-05-00037-f004:**
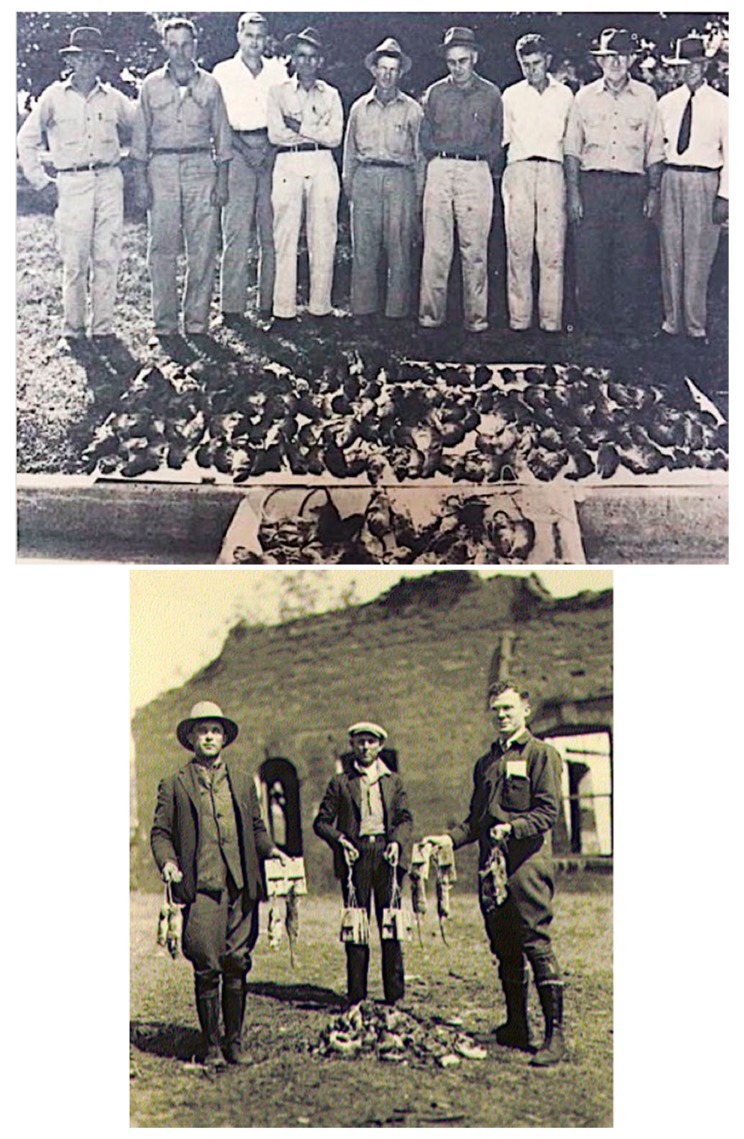
The rat ridders. (**upper**) Texas rat killing squad, 1930s, proudly displaying their quarry [136]. (**lower**) Men holding traps and dead rats in Geneva County, Alabama as part of a rat eradication project of the Civil Works Administration, 22 February 1934.

**Figure 5 tropicalmed-05-00037-f005:**
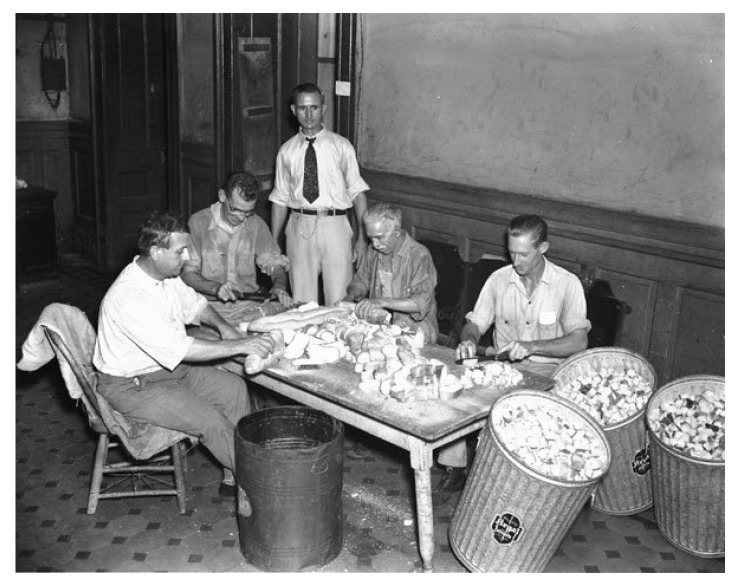
Works Progress Administration workers preparing rodenticide torpedoes (typically a mixture of red squill with either meat, fish, or grain, wrapped in paper), New Orleans, 1936.

**Figure 6 tropicalmed-05-00037-f006:**
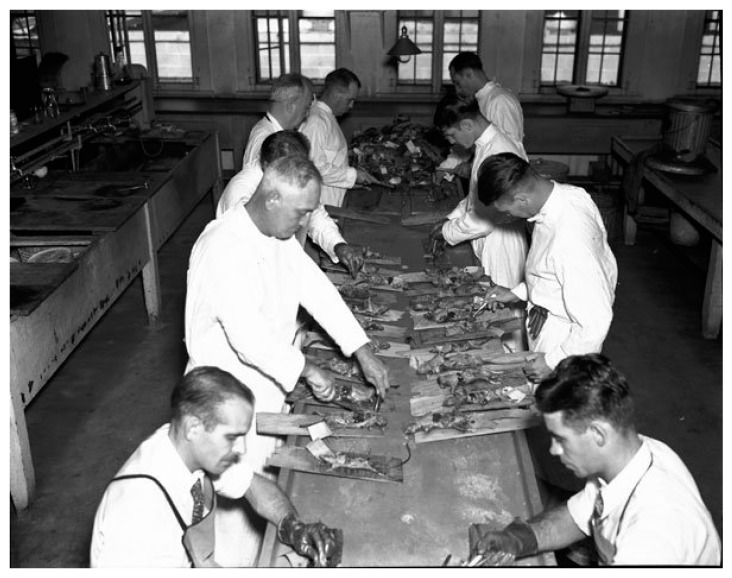
Works Progress Administration workers nailing rats to shingles for inspection, New Orleans, 1936.

**Figure 7 tropicalmed-05-00037-f007:**
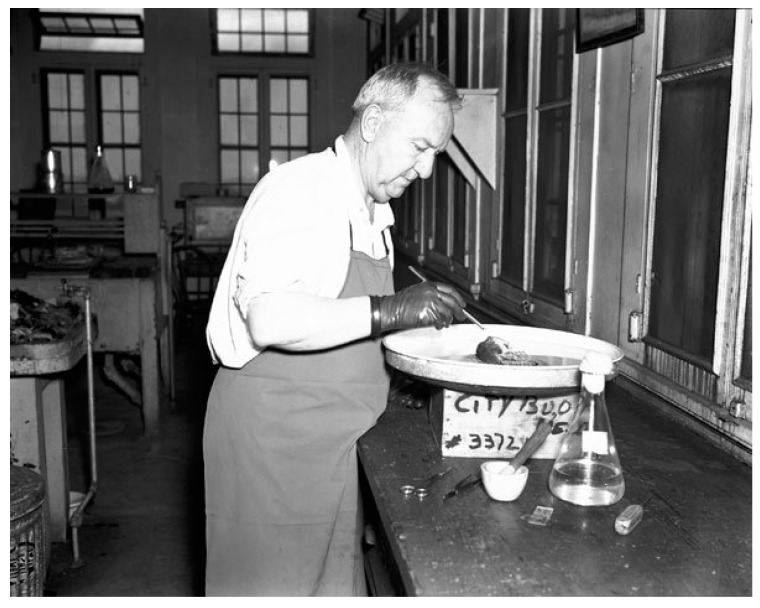
Works Progress Administration technician removing fleas from rat for quantification and identification, New Orleans, 1936.

**Figure 8 tropicalmed-05-00037-f008:**
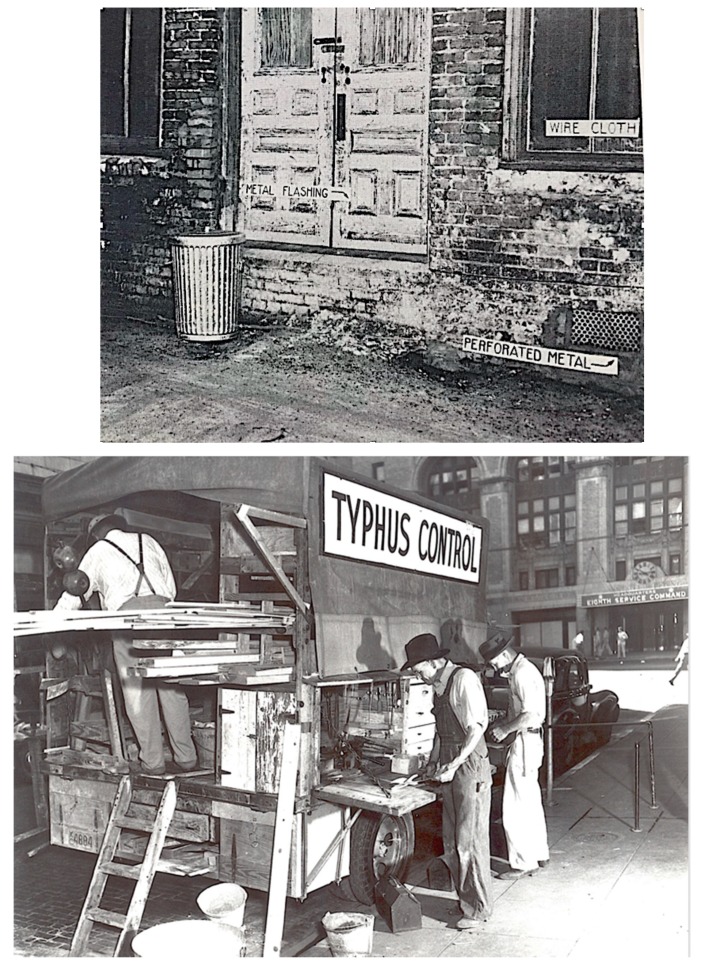
(**upper**) Entrance portals for rats blocked by vent stoppage: metal flashing around edge of a wooden door, screening for a window, and a perforated metal grate [156]. (**lower**) Men involved in vent stoppage against rats in San Antonio, TX, likely late 1930s/early 1940s.

**Figure 9 tropicalmed-05-00037-f009:**
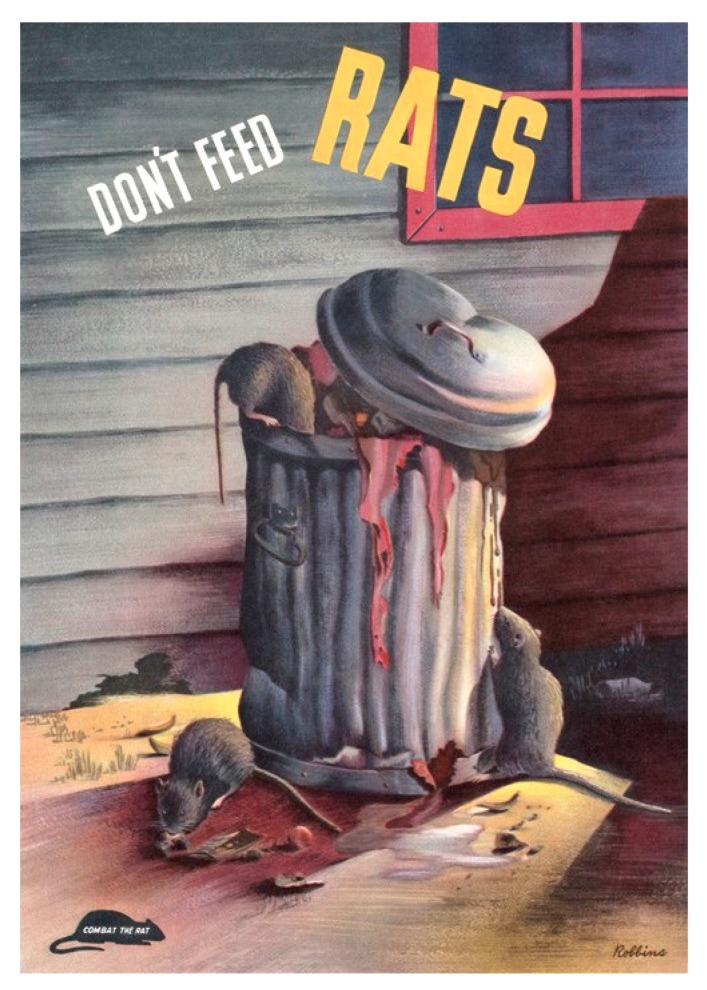
Don’t feed rats. Combat the rat. This US Public Health Service poster from 1944 encourages the proper disposal of food waste.

**Figure 10 tropicalmed-05-00037-f010:**
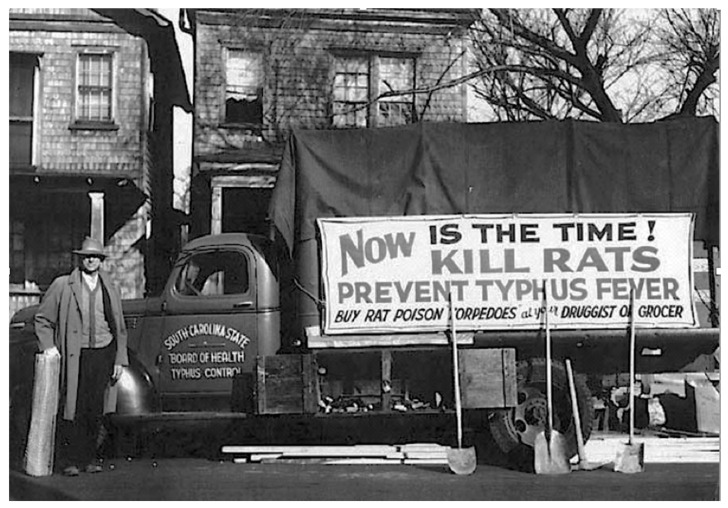
Truck for rat proofing, promoting the use of rat poison torpedoes. Charleston, SC, 1939.

**Figure 11 tropicalmed-05-00037-f011:**
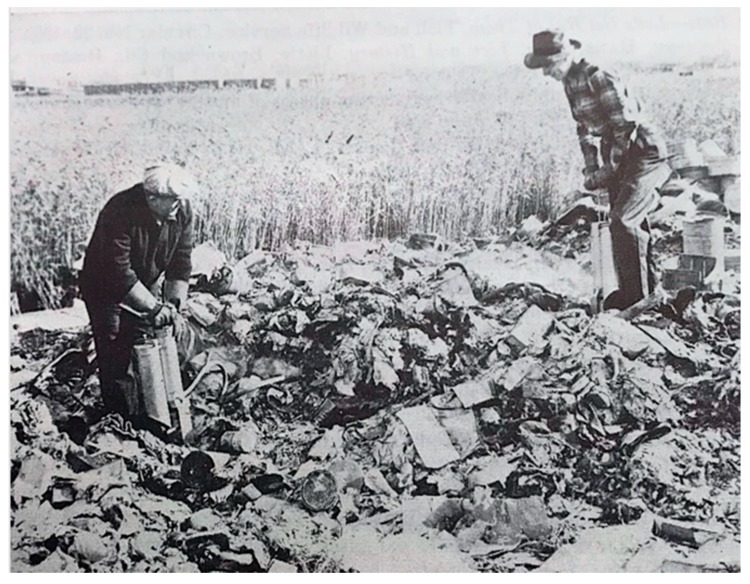
Men pumping calcium cyanide powder (Cyanogas^TM^) into rat burrows at a city dump. This compound reacts with moisture and releases hydrogen cyanide gas [156].

**Figure 12 tropicalmed-05-00037-f012:**
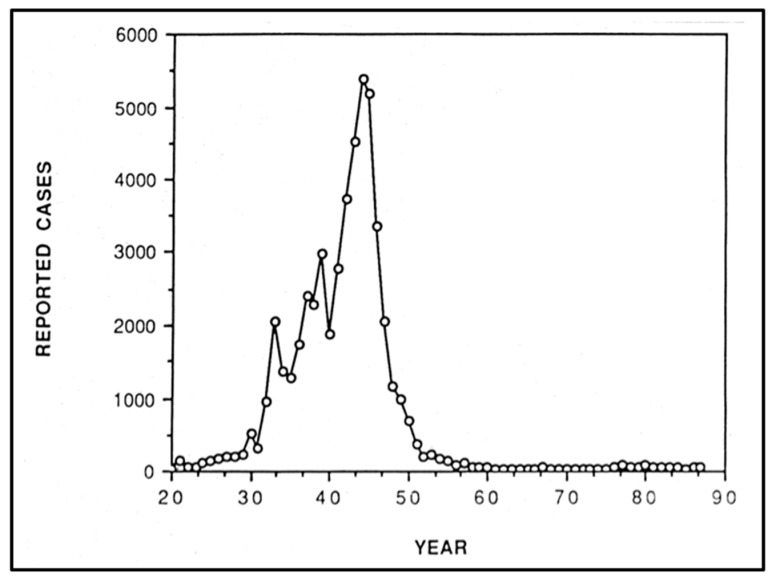
Flea-borne typhus fever cases in the United States, 1920–1987 [1].

**Table 1 tropicalmed-05-00037-t001:** Reported complications of flea-borne typhus.

Complications ^a^
**Neurologic:** aseptic meningitis, hemiparesis, cerebellitis, facial nerve palsies, seizures, ataxia, altered mental status, transient hearing loss, abducens nerve palsy, neurocognitive deficits, cerebral infarction, encephalopathy, intracranial hemorrhage, brain abscess
**Ocular:** oculoglandular syndrome, retinitis, optic neuritis, uveitis, subconjunctival hemorrhage
**Pulmonary:** pneumonia, pulmonary edema, adult respiratory distress syndrome, pulmonary embolism
**Cardiac:** endocarditis, pericarditis, myocarditis, coronary arteritis
**Gastrointestinal:** hepatitis, intestinal pseudo-obstruction, acalculous cholecystitis, pancreatitis
**Hematologic:** hemophagocytic syndrome, hemolysis, bone marrow granulomatosis, splenic infarction, splenic rupture, coagulopathy, venous thrombosis
**Renal:** renal failure
**Other:** parotitis, myositis, suppurative arthritis, leukocytoclastic vasculitis, septic shock, multi-organ failure, death

^a^ Anstead et al., manuscript in preparation.

**Table 2 tropicalmed-05-00037-t002:** Cumulative number of cases and approximate cumulative incidence of flea-borne typhus in the Southeastern states and CA, 1922–1939 ^a,b^.

Rank ^d^	State	Cumulative Number of Cases, 1922–1939	Avg Annual Cases	Avg Pop ^c^ 1920, 1930, 1940	Approx Cumulative Incidence; Cases/ 100,000 Avg Pop	Avg Annual Incidence/ 100,000 Avg Pop
1	GA	6225	345.8	2,976,020	209	11.6
2	AL	3751	208.4	2,609,128	144	8.0
3	TX	3277	182.1	5,634,256	58.1	3.2
4	FL	806	44.8	1,444,698	55.8	3.1
5	SC	707	39.3	1,774,098	39.9	2.2
6	NC	481	26.7	3,100,341	15.5	0.86
7	LA	244	13.6	2,087,994	11.7	0.65
8	MD	195	10.3	1,634,144	11.9	0.63
9	VA	181	10.1	2,531,172	7.51	0.42
10	TN	168	9.3	2,623,427	6.4	0.35
11	MS	158	8.8	1,994,745	7.9	0.44
12	CA	139	7.7	5,337,166	2.6	0.14
	Total	16,332				

^a^ [128]. ^b^ Some reported cases from rural TN, NC, MD, and VA may have been Rocky Mountain spotted fever. ^c^ Average population calculated from average population of 1920, 1930, and 1940, from US Census Bureau Data. ^d^ Rank by cumulative number of cases.

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
