# Peer review of "History, Rats, Fleas, and Opossums: The Ascendency of Flea-Borne Typhus in the United States, 1910–1944"

_tropicalmed, 2020, doi:10.3390/tropicalmed5010037_

Round 1
Reviewer 1 Report
This is quite an interesting paper and generally well written. It is a very long manuscript, more akin to a book chapter.
I have only a few minor comments
Although opossums are in the title, they are really not talked about to any extent in this first chapter. I understand the homage aspect of the title- but maybe include a forewarning in the introduction that the role of the opossums will not be discussed in this paper if the authors choose not to modify the title.
Line 68, remove the hyphen between 14 and days
a short description of the rash should be provided
line 76, put an "a" before Triad
for table 1 you have used your own unpublished manuscript as a reference for the complications. It would be preferable to have another reference source that is published for this list of complications.
Author Response
Although opossums are in the title, they are really not talked about to any extent in this first chapter. I understand the homage aspect of the title- but maybe include a forewarning in the introduction that the role of the opossums will not be discussed in this paper if the authors choose not to modify the title.
OK, the role of opossums is now clearly delineated in the introduction.
"Opossums are mentioned in the title, but they do not play a significant role in this initial time period; their importance will be revealed in the susequent parts. Part II (the Decline) will describe: the innovations in insecticide and rodenticide technology occurring during and after World War II; the public health programs instituted for typhus and rodent control in the post-war period; and the improvement in social conditions in the endemic areas after World War II. Part III (the Resurgence) will address the change in the epidemiology of FBT from a rodent flea-borne disease to a cat flea-borne disease, with opossums as a major reservoir; the discovery of a second causative bacterium (R. felis); and epidemiologic trends in the persistent FBT endemic areas of the USA (TX, CA, and HI)."
Line 68, remove the hyphen between 14 and days
Ok, done.
a short description of the rash should be provided.
Ok, rash now described.
In one series of eighty patients, rash occurred in 54%; the erythematous rash was macular in 49%, maculopapular in 29%, papular in 14%, petechial in 6%, and morbilliform in 3% [8].
line 76, put an "a" before Triad
Ok, done.
for table 1 you have used your own unpublished manuscript as a reference for the complications. It would be preferable to have another reference source that is published for this list of complications.
There really is no comprehensive summary paper of the multiple complications of flea-borne typhus,which is why I am working on a review for this topic. More than 40 references would be necessary to document all the complications that are listed in the table. I request that the reviewer and editor allow the table due to the large number of references already in the manuscript.
Reviewer 2 Report
Very well written, exhaustive review on the history of flea-borne typhus in the US. I found it to be highly informative and easy to read. Other than several small editorial issues, I recommend this article be published in its current state.
Line 126: Chicken seems to be either in a different font or size.
Line 264: Font or spacing seems to have been changed or is smaller.
Figure 4 & all figures in general, it seems like there is variability in the font size of figure legends.
Author Response
Line 126: Chicken seems to be either in a different font or size.
Will adjust as requested
Line 264: Font or spacing seems to have been changed or is smaller.
Will adjust as requested
Figure 4 & all figures in general, it seems like there is variability in the font size of figure legends.
Will adjust as requested